# Transcriptome-Wide Analysis of Human Chondrocyte Expansion on Synoviocyte Matrix

**DOI:** 10.3390/cells8020085

**Published:** 2019-01-24

**Authors:** Thomas J. Kean, Zhongqi Ge, Yumei Li, Rui Chen, James E. Dennis

**Affiliations:** 1Orthopedic Surgery, Baylor College of Medicine, Houston, TX 77030, USA; James.Dennis@bcm.edu; 2Department of Molecular and Human Genetics, Baylor College of Medicine, Houston, TX 77030, USA; zge@mdanderson.org (Z.G.); yumeil@bcm.edu (Y.L.); ruichen@bcm.edu (R.C.)

**Keywords:** chondrocyte RNA-Seq, dedifferentiation, chondrogenesis, synoviocyte matrix, physioxia, RNA-Seq, cell senescence

## Abstract

Human chondrocytes are expanded and used in autologous chondrocyte implantation techniques and are known to rapidly de-differentiate in culture. These chondrocytes, when cultured on tissue culture plastic (TCP), undergo both phenotypical and morphological changes and quickly lose the ability to re-differentiate to produce hyaline-like matrix. Growth on synoviocyte-derived extracellular matrix (SDECM) reduces this de-differentiation, allowing for more than twice the number of population doublings (PD) whilst retaining chondrogenic capacity. The goal of this study was to apply RNA sequencing (RNA-Seq) analysis to examine the differences between TCP-expanded and SDECM-expanded human chondrocytes. Human chondrocytes from three donors were thawed from primary stocks and cultured on TCP flasks or on SDECM-coated flasks at physiological oxygen tension (5%) for 4 passages. During log expansion, RNA was extracted from the cell layer (70–90% confluence) at passages 1 and 4. Total RNA was column-purified and DNAse-treated before quality control analysis and next-generation RNA sequencing. Significant effects on gene expression were observed due to both culture surface and passage number. These results offer insight into the mechanism of how SDECM provides a more chondrogenesis-preserving environment for cell expansion, the transcriptome-wide changes that occur with culture, and potential mechanisms for further enhancement of chondrogenesis-preserving growth.

## 1. Introduction

Arthritis is not only a debilitating disease, but an expensive one, with total arthritis-attributable medical expenditures and lost earnings surpassing $300 billion USD in 2013 [1]. Tissue engineering methods have been applied as a means to treat osteoarthritic lesions and they hold great potential for joint repair. However, human chondrocytes, expanded and used in autologous chondrocyte implantation techniques, are known to rapidly de-differentiate in culture [2], which has a detrimental impact on their utility for tissue engineering applications. The use of the term de-differentiation is distinct from the use of the term in re-programming or stem cell literature, as it indicates that the cells no longer have the ability to form hyaline-like cartilage tissue. Culture-expanded chondrocytes undergo both morphological and phenotypical changes and, eventually, lose the ability to produce hyaline like matrix. By passage 4, chondrogenic potential is essentially absent when tested in re-differentiation culture. This loss of differentiation potential limits their efficacy in the clinic and seriously impedes our ability to produce clinical-scale tissue engineering of human cartilage with suitable biomechanical properties. 

Synoviocyte-derived extracellular matrix has been shown to support enhanced chondrocyte expansion whilst retaining re-differentiation potential in human [3] and porcine cells [4]. Next-generation sequencing offers an opportunity to take a global view of the transcriptional changes that occur during in vitro culture both on tissue culture plastic and on devitalized synoviocyte matrix. Relatively few RNA sequencing (RNA-Seq) studies have been conducted on human chondrocytes partially due to the dense extracellular matrix they form, making RNA isolation problematic in terms of both yield and quality. No RNA-Seq reports on the effect of synoviocyte matrix or the effect of culture de-differentiation were found. This study is an expansion of previous work [3] which showed typical chondrocyte-related gene expression changes by RT-qPCR (*COL1A1, COL2A1, SOX9, ACAN, MATN1, MMP13, COL10A1*) the levels of which, by fragments per kilobase of exon model per million fragments mapped reads (FPKM), are well correlated. In addition, RNA-Seq has been shown to be well correlated with RT-qPCR data across over 15,300 genes [5]. Preliminary analysis of the effect of passage on the transcriptome of chondrocytes grown on tissue culture plastic (TCP) was presented [6]. The results from this study give us a deeper understanding of chondrocyte biology, thus providing a better foundation for future therapeutic interventions.

## 2. Materials and Methods

Human chondrocytes (three donors) were thawed from frozen, end of primary culture stocks collected under an institutional review board (IRB)-approved protocol (H-36683) of the IRB for Baylor College of Medicine and Affiliated Hospitals. Chondrocytes were cultured on TCP flasks and synoviocyte-derived extracellular matrix (SDECM) at physiological oxygen tension (5%) for 4 passages (Figure 1) in growth media (DMEM-LG [HyClone, Pittsburgh, PA, USA] supplemented with 10% FBS [Atlanta Biologicals, Flowery Branch, GA, USA] and 1% penicillin/streptomycin [Gibco, Gaithersburg, MD, USA]). At the end of the first and fourth passage, cells were lysed and RNA extracted from the cell layer during the log expansion (70–90% confluence) phase. Cell lysis was performed using a guanidine chloride-based buffer (TRK Lysis buffer; (E.Z.N.A.^®^ Tissue RNA Kit, Omega Bio-Tek, Norcross, GA, USA) and the lysate was frozen on dry ice and stored (−80 °C, 1–6 weeks). Companion flasks that were not lysed were trypsinized (0.25% trypsin/EDTA; Gibco, Gaithersburg, MD, USA) for 5 min at 37 °C, then trypsin-neutralized with an equal volume of growth media. Cells were collected and centrifuged (500× *g*, 5 min, room temperature). Cell pellets were resuspended and an aliquot counted using a hemocytometer with trypan blue (1:1; Gibco). Cells were seeded at 6000 cells/cm^2^ and media were exchanged on day 2–3 and the cells cultured for 5–6 days at each passage. 

When all flasks had been lysed, total RNA was isolated from the lysate after thawing on ice using column purification (Direct-zol RNA mini-prep, Zymo Research, Irvine, CA, USA), as per manufacturer’s instructions, with on-column DNA digest (DNase I, Zymo Research). RNA Purity was analyzed by a 260 nm/280 nm ratio (Tecan Nanoquant, Morrisville, NC, USA) and degradation/quality assessed using an Agilent Bioanalyzer (Genomic and RNA Profiling Core at Baylor College of Medicine, Houston, TX, USA). Samples were then submitted for next-generation RNA sequencing. mRNAs were captured by oligo-dT magnetic beads and fragmented. First-strand cDNA was generated using random primers, and second-strand cDNA was synthesized with deoxyuridine triphosphate (dUTP). The library was generated using the ds cDNA as a template. Briefly, templates were end-repaired, and a 3′A was added before Y-shaped adaptors were added to each end. The strand with dUTP was then digested using uracil-DNA glycosylase prior to PCR. Sequencing was carried out on Illumina HiSeq 2000. RNA-Seq reads were mapped to human genome hg19 and splice junction sites with Bowtie (v0.12.7) [7] and Tophat (v2.0.0) [8]. Read counts mapped to each gene were calculated by HTseq [9], and fragments per kilobase of exon model per million fragments mapped reads (FPKM) values were calculated using Cufflinks (version 2.1.1; http://cole-trapnell-lab.github.io/cufflinks/releases/v2.1.1/) [10]. Differential expression was analyzed with R (http://www.R-project.org) [11] and the Bioconductor R package DESeq [12]. Heatmaps were generated from FPKM values with heatmap3. Genes with an adjusted *p*-value less than 0.01 and greater than a 2log2 (4-fold change) expression were considered to be differentially expressed. Network analysis was performed using STRING (http://string-db.org) [13]. Gene Ontology (GO) term enrichment was queried using a logically accelerated GO term finder (LAGO) [14]. Common genes were determined by combining table queries and uncommon genes by unmatched queries in Microsoft Access (v14.0.7214.5000, Redmond, WA, USA). GO Term lists were downloaded from Jax [15].

RNA-Seq data was deposited in the SRA database under accession number SRP156000.

## 3. Results

Chondrocytes grown on TCP typically expanded for 1.8 population doublings (PD) in P1 and 1.4 PD at P4, while those grown on SDECM expanded 3.6 PD in P1 and 4.0 PD at P4 [3]. At the end of P4, cells grown on TCP had undergone 7.6 PDs and those on SDECM had undergone 16.0 PDs. RNA was of high quality with RNA integrity numbers (RINs) ≥7.6 (See Appendix A for electropherograms and Appendix A for summary). Principal component analysis showed that donors clustered by both passage (P1 vs. P4) and surface (SDECM vs. TCP; Figure 2). 

Many genes were significantly affected by passage and culture surface, an overview of the up and down regulated genes for each comparison is shown in Table 1. Of the genes that were differentially regulated between P1 and P4, 512 were common between the two comparisons with 228 genes being upregulated at P4 and 283 genes downregulated. Only lipase G (*LIPG*) switched direction in cells cultured on the two different surfaces, being downregulated at P4 on SDECM and upregulated on TCP. Looking at the genes that were not common between the two surfaces, 175 were upregulated on TCP and 337 were downregulated. The upregulated genes in this subset were enriched for terms such as “tissue development” and “multicellular development”. The downregulated genes were enriched for terms like “skeletal development”, “system development” and “extracellular region”. Of the 251 genes on SDECM that were not common between the two comparisons, 88 genes were upregulated and 163 downregulated. The upregulated genes in this subset were not enriched for any biological process, cellular component or molecular function. The downregulated genes in this subset were enriched for sterol-, cholesterol-, and lipid-related terms.

The top 20 genes for each comparison, in terms of adjusted *p*-value, are shown in Table 2 and Table 3. Full lists of differentially expressed genes, the gene count output from HTseq, the FPKM output from Cufflinks, and the results of DESeq comparisons are shown in Appendix A.

In comparisons 1 and 2, the major enriched pathways by GO analysis were cell cycle-associated. This was primarily due to the upregulated genes and resulted in an increase in both the G1/S and G2/M cell cycle checkpoint genes (Figure 3 and Figure 4). Downregulated genes were enriched for GO terms like “system development” and “multicellular organism development”. Genes were tallied from GO terms associated with positive or negative regulation of the cell cycle; there is an increase in the number of genes negatively regulating cell proliferation on TCP (140 genes vs. 51 on SDECM). However, caution should be exercised in using these GO term lists alone, as 44 of the 182 genes were present in both positve and negative terms. There were 35 genes associated only with positive regulation of proliferation on SDECM at P4 and 23 on TCP; 25 genes were only associated with negative regulation of proliferation on SDECM at P4 and 95 on TCP (tabulated data are included in S5).

The GO term for “extracellular matrix organization” (GO: 0030198) is associated with 268 genes. When the differentially expressed genes from comparisons 1, 2, 3, and 4 were queried with this list, 50 genes were identified as being differentially expressed in one or more comparison (Figure 5). Comparison 1 showed differential expression of 30 genes, with eight of those genes being upregulated at P4 on TCP, and 22 downregulated at P4 on TCP. Comparison 2 identified 22 differentially expressed genes, 4 of which were upregulated at P4 on SDECM and 18 downregulated at P4 on SDECM. Extracellular matrix disassembly (GO: 0022617) identified 29 genes, of which only three genes were differentially regulated in any of the comparisons: matrix metalloproteinase 13 (*MMP13*), semaphorin 5A (*SEMA5A*) and fibroblast growth factor receptor 4 (*FGFR4*). *MMP13* decreased with passage on both surfaces but was increased at both P1 and P4 by culture on SDECM. *SEMA5A* was increased at P1 by culture on SDECM. *FGFR4* significantly decreased with passage on SDECM but was increased on SDECM in comparison with TCP at P1. 

Of the 204 genes that are identified by the gene ontology term “cartilage development” (GO:0051216), 43 genes were differentially expressed in one or more comparisons. At P4 on TCP, 28 genes were significantly decreased vs. 16 on SDECM at P4; only four were upregulated on TCP and three on SDECM (Figure 6). 

Looking at genes associated with the GO Term “cell senescence” (GO:0090398, 63 genes), nine were differentially regulated in comparison 1: *BCL6, C2orf40, NOX4, SERPINE1, BRCA2, FBXO5, SUV39H1, CDK1,* and *FOXM1*. Four of these genes were also differentially regulated in comparison 2: *NOX4, CDK1, FOXM1,* and *C2orf40*.

*COL2A1* is a well-known marker for hyaline cartilage, and since it was decreased in all four comparisons, transcription factors regulating its expression were further investigated. Only *BCL6* and *PPARG* were significantly downregulated ≥4-fold on TCP, and *PPARG* alone on SDECM. The conversion from type II collagen expression to type I collagen expression is a distinctive marker for fibrocartilage vs. hyaline cartilage [16,17]. While *COL1A1* is not one of those genes which met the 4-fold increase selection criterion, it was still increased 2.8-fold on TCP with an adjusted *p*-value of 1 × 10^−30^; the upregulation on SDECM was 1.2-fold and non-significant. Out of the over 300 transcription factors with an identified (or potential) binding sequence in the *COL1A1* gene [18] only five were significantly upregulated on TCP and 4 on SDECM Table 4.

When considering the effect of culture surface, in comparison 3, the most significantly enriched GO terms related to extracellular matrix (14 of 115 GO terms, *p* < 0.01; S5). Downregulated genes were also significantly enriched for “extracellular matrix” and “cell motility” GO Terms (S6). Upregulated genes were enriched for GO Terms related to development and differentiation (S7).

In comparison 4, the most significantly enriched GO terms related to “development” and “extracellular matrix”. “Extracellular matrix” terms dominated the enriched GO term list for downregulated genes (S8) whilst neuron-related terms predominated in the upregulated GO term list (S5, S9). Of the 339 differentially expressed genes at P4, 107 were common with P1; 52 went up and 55 went down. Two genes switched their direction: angiopoietin-like 7 (*ANGPTL7*) went from being downregulated on SDECM at P1 to being upregulated on SDECM at P4; laminin subunit gamma 2 (*LAMC2*) was upregulated on SDECM at P1 and flipped to being downregulated at P4. GO Term analysis of the downregulated genes recapitulated the predominance of terms for “extracellular matrix” as did the upregulated genes for neuronal terms. Differentially expressed genes at P1 that were not common between the two surface comparisons: 224 genes, 99 were upregulated on SDECM and 125 were downregulated. Upregulated genes were enriched in GO terms for extracellular matrix organization and vasculature development. Downregulated genes were enriched in GO terms for “extracellular region”. Differentially expressed genes at P4 that were not common between the two surface comparisons: 232 genes, 110 were upregulated on SDECM and 122 were downregulated. Upregulated genes were enriched in GO terms for “multicellular” and “system development”. Downregulated genes were enriched in cellular component GO terms for “extracellular region”. Selection of genes that were unchanged on SDECM between P1 and P4 (less than a 10% increase or decrease), but that were significantly differentially expressed on TCP gave 15 genes (Table 5).

## 4. Discussion

The earliest RNA-Seq study on chondrocytes appears to be that by Peffers et al. [19] comparing young and old horse cartilage. Peffers et al. found 396 genes changed more than 2.6-fold with a *p*-value <0.05; a relatively smaller differential gene expression than that reported in this study. In fact, the large number of differentially expressed genes prompted the more stringent *p*-value and higher fold-change cut offs. There was a large degree of correlation between comparisons 1 and 2, with half of the top 20 genes being differentially regulated. Overall, half of the genes were common from the TCP comparison and two-thirds of those from the SDECM comparison. The coincidence of the differential regulation of two lipid-related genes—lipase G (*LIPG*) and peroxisome proliferator activated receptor gamma (*PPARG*)—is interesting particularly since *PPARG* stimulation by Rosiglitazone has been shown to be chondroprotective [19]. This chondroprotection potentially acts through a blunting of NF-kappa-B-mediated inflammatory signals [20]. However, in a more recent study, Xu et al. found that *PPARG* stimulation was detrimental to the expression of *COL2A1* and promoted hypertrophy [21]. It is worth noting that the transcription factors *SOX9* and *SOX6*, commonly thought essential for chondrogenesis, were not significantly downregulated on SDECM at P4; this is potentially a major contributor to the perpetuation of chondrogenicity found in cells cultured on synoviocyte matrix. Interestingly, *SOX5*, another putative essential cartilage transcription factor [22,23], is 3.4-fold downregulated at P4 on TCP (Padj 7.9 × 10^−11^) vs. 1.7-fold downregulated at P4 on SDECM (Padj 6.4 × 10^−4^), and *SOX5* was upregulated in both comparisons 3 and 4 (1.4-fold and 2.8-fold respectively). 

Whilst immortalized hepatocytes have been shown to increase hepatocyte markers when cultured on soft substrates [24], this has not been shown for de-differentiated primary human chondrocytes. An increase in chondrogenic markers is often achieved in cell culture expanded cells [25,26]. However, the expression of these markers is commonly deficient when compared to non-expanded chondrocytes. Also, soft substrates do not promote, but inhibit expansion [27]. In the case of hepatocytes and b-cells, de-differentiation has been linked to epithelial–mesenchymal transition [28]; this would not be the case for chondrocytes, as stimulation of transforming growth factor beta (TGFβ) pathways, rather than inhibition, promotes chondrogenesis.

The increase in cell cycle control genes could be indicative of cell culture adaptation, as this was described by Barta et al. [29] who showed increased cyclin E and cyclin A in human embryonic stem cells cultured for over 240 passages; both cyclins were increased at passage 4 (3.2-fold and 9.1-fold in comparisons 1 and 2, respectively). This potentially futile growth, a conversion from reversible cell cycle arrest to irreversible senescence, could be indicative of geroconversion as described by Blagosklonny [30,31,32]. Interestingly, by blocking mTOR signaling using rapamycin, several studies have shown a positive impact on chondrogenesis and in osteoarthritis [33,34,35]. When we looked at cell senescence-associated genes several were differentially regulated at higher passage on both surfaces. Those genes, *NOX4, CDK1, FOXM1*, and *C2orf40*, represent interesting targets for modulation to further enhance the retention of chondrogenic capacity found on synoviocyte matrix. F-box protein 5 (*FBXO5*) is notable because of its presence in the cells passaged on TCP (comparison 1, Figure 3) and its absence on SDECM (comparison 2, Figure 4). FBXO5 is part of the Skp, Cullin, F-box containing complex that catalyzes the ubiquitination of proteins marking them for degradation. Increased FBXO5 could result in more cells being held at the G1/S and G2/M checkpoints but warrants more study. The increased number of genes promoting proliferation on SDECM vs. TCP (Figure 3 and Figure 4), and perhaps more importantly, the ratio of genes promoting vs. inhibiting proliferation is thought to be partially responsible for the increased proliferation of chondrocytes on SDECM.

Comparisons 3 and 4 resulted in upregulation of development-related genes and down regulation of matrix-associated genes and those involved in motility. It is postulated that, because there is an extracellular matrix already present, the chondrocytes are less invested in laying down a new matrix. Chondrocytes in their native environment are relatively immobile though in vitro motility has been demonstrated by several studies [36]. The potential reduction in movement on SDECM could indicate that the chondrocytes are in a more “native” state and is another interesting avenue for investigation. 

Genes that were stable on SDECM but significantly changed on TCP were identified, as this represents the pool of genes that could be targeted for modulation to recapitulate the effects of growth on SDECM (Table 5). Network and gene enrichment analyses of the genes in Table 4 showed no overall connection between them. This is potentially a consequence of the dearth of studies on the musculoskeletal transcriptome, particularly in cartilage, whereas many cancer studies have contributed to the gene ontology terms and their association. It should also be noted that chondrocyte proliferation, studied here, appears to be a somewhat separate process from chondrogenesis or cartilage development in terms of differentiation markers.

## 5. Conclusions

Chondrocyte growth on devitalized synoviocyte matrix dramatically changes the transcriptomic signature of the cells, predominantly in extracellular matrix-associated genes and those related to cell motility. De-differentiation due to passage of chondrocytes also dramatically alters the transcriptome, predominantly resulting in cell cycle gene expression changes.

## Figures and Tables

**Figure 1 cells-08-00085-f001:**
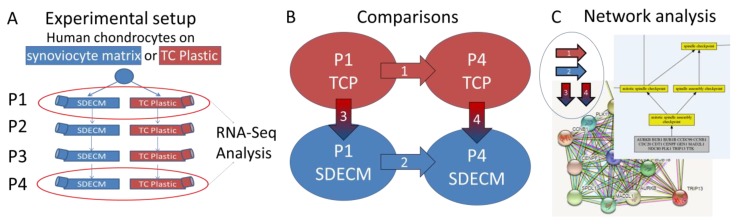
Experimental outline and comparisons. (**A**) Experimental setup: human articular chondrocytes were thawed from frozen stocks and seeded onto both synoviocyte-derived extracellular matrix (SDECM) and tissue culture plastic (TCP) flasks, passaged 4 times and RNA collected at passage (P) P1 and P4 for RNA sequencing (RNA-Seq) analysis. (**B**) Comparisons of gene expression profiles (1) P1 vs. P4 on TCP, (2) P1 vs. P4 on SDECM, (3) TCP vs. SDECM at P1, (4) TCP vs. SDECM at P4. (**C**) Network and gene enrichment analyses of differentially expressed genes from each comparison.

**Figure 2 cells-08-00085-f002:**
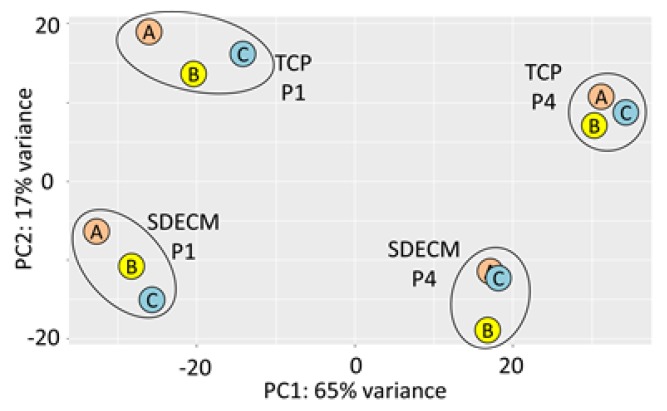
Principal component analysis. Samples are clearly clustered by their passage (P) in principal component (PC) PC1 and surface in PC2; donors are indicated by the letter within the circle (A, B, C).

**Figure 3 cells-08-00085-f003:**
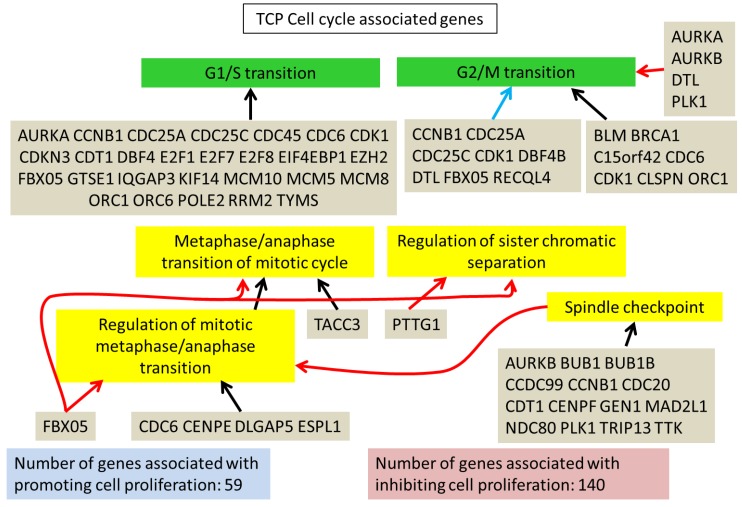
TCP cell cycle-associated genes. A summary of some of the cell cycle-associated genes from LAGO analysis which were upregulated in P4 chondrocytes on TCP. Similar cell cycle enrichment was seen in upregulated genes at P4 on SDECM (Figure 4). Red arrows connecting the genes to a term indicate inhibition of that term, blue arrows = promotion and black arrows = association. For the full interaction chart see S3 (TCP) and for the list of Gene Ontology (GO) terms which were significantly enriched see S5.

**Figure 4 cells-08-00085-f004:**
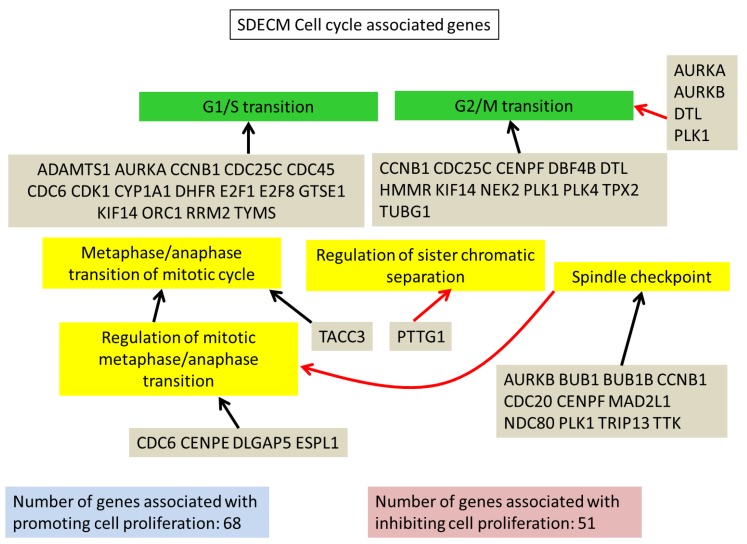
SDECM cell cycle-associated genes. A summary of some of the cell cycle-associated genes from LAGO analysis which were upregulated in P4 chondrocytes on SDECM. Red arrows connecting the genes to a term indicate inhibition of that term, blue arrows = promotion and black arrows = association. For the full interaction chart see S4 (SDECM) and for the list of GO terms which were significantly enriched see S5.

**Figure 5 cells-08-00085-f005:**
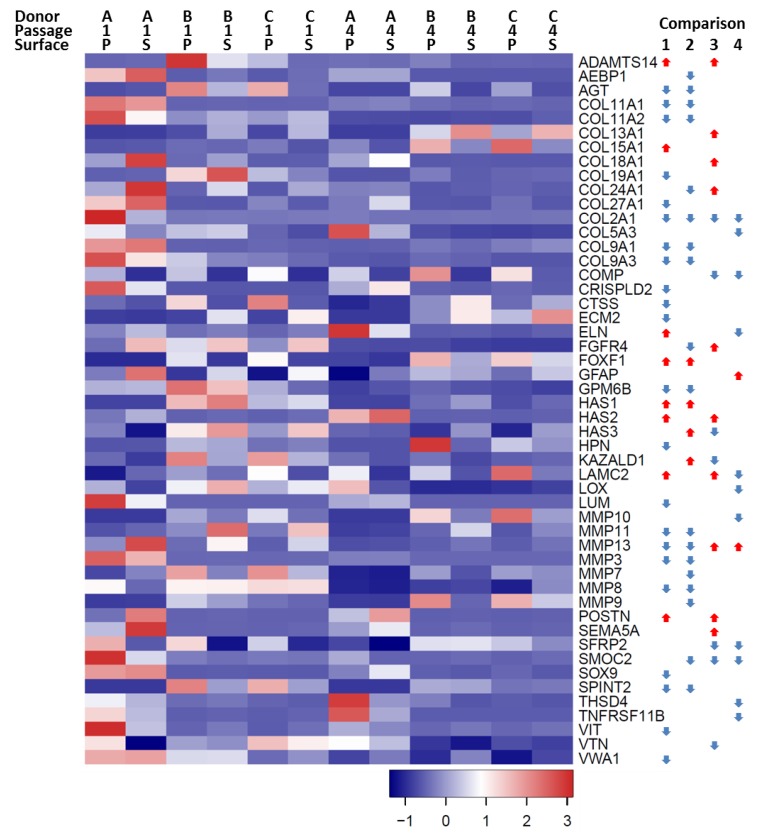
Differentially expressed extracellular matrix organization genes. This heatmap shows differentially expressed genes from the four comparisons which are associated with the GO term “extracellular matrix organization”. Donor (A, B, C), passage (1, 4) and surface (P = TCP and S = SDECM) are indicated at the top of the heatmap. The gene symbol is to the right of the heatmap. The arrows indicate which comparisons were significantly different at a ≥4-fold change with *p* < 0.01; no arrow means that the comparison did not meet that threshold (1 = TCP P1 vs. P4, 2 = SDECM P1 vs. P4, 3 = TCP vs. SDECM at P1, 4 = TCP vs. SDECM at P4). A red arrow pointing upwards (
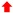
) indicates that gene expression increased and a blue downward-pointing arrow (
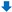
) that it decreased.

**Figure 6 cells-08-00085-f006:**
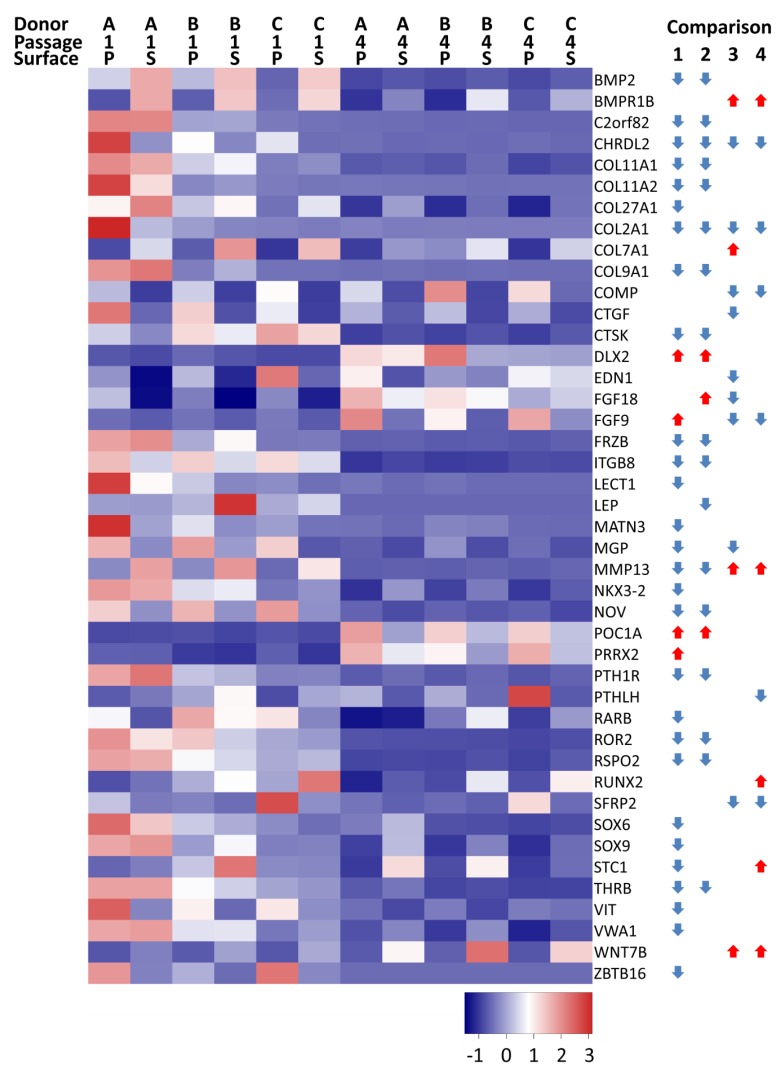
Differentially expressed cartilage development genes. Heatmap summary of differentially expressed genes associated with the GO term “cartilage development”. Donor (A, B, C), passage (1, 4), and surface (P = TCP and S = SDECM) are indicated at the top of the heatmap. The gene symbol is to the right of the heatmap. The arrows indicate which comparisons were significantly different at a ≥4-fold change with *p* < 0.01; no arrow means that the comparison did not meet that threshold (1 = TCP P1 vs. P4, 2 = SDECM P1 vs. P4, 3 = TCP vs. SDECM at P1, 4 = TCP vs. SDECM at P4). A red arrow pointing upwards (
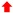
) indicates that gene expression increased and a blue downward-pointing arrow (
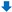
) that it decreased.

**Table 1 cells-08-00085-t001:** Summary of differentially expressed genes (Adjusted *p*-value < 0.01 and log2 fold-change >2).

Comparison	Upregulated (%)	Downregulated (%)	Total	Common ^3^
(1) P1 vs. P4 on TCP ^1^	404 (39)	620 (61)	1024	512
(2) P1 vs. P4 on SDECM ^1^	316 (41)	447 (59)	763	
(3) TCP vs. SDECM P1 ^2^	151 (46)	180 (54)	331	107
(4) TCP vs. SDECM P4 ^2^	162 (48)	177 (52)	339	

^1^ Up/downregulated at P4. ^2^ Up/downregulated on SDECM. ^3^ Genes that were differentially expressed in both comparisons 1 and 2 or comparisons 3 and 4.

**Table 2 cells-08-00085-t002:** Top 20 differentially expressed genes in comparisons 1 and 2.

Comparison 1			Comparison 2		
Gene Symbol	Gene Name	FC ^1^	Gene Symbol	Gene Name	FC ^1^
*MAFB*	MAF BZIP transcription factor B	25.5 D	*APLN*	Apelin	21.8 D
*PLK1*	Polo-like kinase 1	23.1 U	*MMP1*	Matrix metallopeptidase 1	26.6 D
*CENPF*	Centromere protein F	12.0 U	*KIF20A*	Kinesin family member 20A	27.6 U
*SLC40A1*	Solute carrier family 40 member 1	166. D	*TOP2A*	DNA topoisomerase II alpha	16.3 U
*TOP2A*	DNA topoisomerase II alpha	13.0 U	*CDC20*	Cell division cycle 20	21.1 U
*MKI67*	Marker of proliferation Ki-67	14.5 U	*TPX2*	TPX2, microtubule nucleation factor	16.0 U
*CCNB1*	Cyclin B1	15.0 U	*FOXM1*	Forkhead box M1	12.3 U
*FGFR2*	Fibroblast growth factor receptor 2	19.0 D	*BIRC5*	Baculoviral IAP repeat containing 5	19.8 U
*ISM1*	Isthmin 1	25.5 D	*MKI67*	Marker of proliferation Ki-67	21.4 U
*CDC20*	Cell division cycle 20	19.6 U	*PRC1*	Protein regulator of cytokinesis 1	12.6 U
*PRC1*	Protein regulator of cytokinesis 1	12.1 U	*DLGAP5*	DL-associated protein 5	23.6 U
*A2M*	Alpha-2-macroglobulin	21.9 D	*PLK1*	Polo-like kinase 1	17.7 U
*HMMR*	Hyaluronan-mediated motility receptor	20.9 U	*SLC40A1*	Solute carrier family 40 member 1	25.9 D
*DLGAP5*	DLG-associated protein 5	17.1 U	*ASPM*	Abnormal spindle microtubule assembly	10.9 U
*AURKA*	Aurora kinase A	20.0 U	*CCNB1*	Cyclin B1	11.7 U
*CCNB2*	Cyclin B2	17.2 U	*PRR11*	Proline rich 11	9.43 U
*TPX2*	TPX2, microtubule nucleation factor	14.2 U	*CENPF*	Centromere protein F	12.6 U
*STEAP4*	STEAP4 metalloreductase	14.1 D	*KIF23*	Kinesin family member 23	13.6 U
*ELN*	Elastin	7.66 U	*CEP55*	Centrosomal protein 55	14.6 U
*PRR11*	Proline rich 11	11.9 U	*ANLN*	Anillin actin binding protein	12.2 U

^1^ Fold change shows the up (U)- or down (D)- regulated fold change of the respective gene in P4. Adjusted *p*-values were ≤ 2.4 × 10^−52^. Genes highlighted with the same colour are common between the two comparisons.

**Table 3 cells-08-00085-t003:** Top 20 differentially expressed genes in comparisons 3 and 4.

Comparison 3			Comparison 4		
Gene Symbol	Gene Name	FC ^1^	Gene Symbol	Gene Name	FC ^1^
*POSTN*	Periostin	16.9 U	*COLEC12*	Collectin subfamily member 12	32.3 U
*COMP*	Cartilage oligomeric matrix protein	28.5 D	*MME*	Membrane metalloendopeptidase	9.11 U
CLEC3B	C-type lectin domain family 3 member B	14.7 D	*CRLF1*	Cytokine receptor-like factor 1	7.62 D
DCN	Decorin	5.87 D	*MFAP5*	Tetratricopeptide repeat domain 9	14.5 D
PODN	Podocan	8.89 D	*TTC9*	Microfibril-associated protein 5	12.1 D
CTGF	Connective tissue growth factor	5.52 D	*IGFBP1*	Insulin-like growth factor binding protein 1	11.9 D
*ADAMTS5*	ADAM metallopeptidase thrombospondin type 1 motif 5	9.99 D	*CLEC3B*	C-type lectin domain family 3 member B	6.48 D
*PPAP2B*	Phospholipid phosphatase 3	5.57 D	*DSP*	Desmoplakin	10.1 D
*TAGLN*	Transgelin	7.23 D	*LOX*	Lysyl oxidase	4.02 D
*SFRP4*	Secreted frizzled-related protein 4	9.26 D	*IGFBP5*	Insulin-like growth factor binding protein 5	4.61 D
*CAMK2N1*	Calcium/calmodulin-dependent protein kinase II inhibitor 1	11.3 U	*C6orf132*	Chromosome 6 open reading frame 132	9.53 D
*FHL1*	Four and a half LIM domains 1	5.66 D	*MOXD1*	Monooxygenase DBH-like 1	7.35 U
*OMD*	Osteomodulin	4.92 D	*CAMK2N1*	Calcium/calmodulin-dependent protein kinase II inhibitor 1	20.1 U
*COL18A1*	Collagen type XVIII alpha 1 chain	6.22 U	*PHLDA1*	Pleckstrin homology-like domain family A Member 1	4.67 U
*SEMA5A*	Semaphorin 5A	4.22 U	*EMB*	Embigin	6.48 U
*SCD*	Stearoyl-CoA desaturase	5.78 U	*PITPNM3*	PITPNM family member 3	4.76 D
*MYL9*	Myosin light chain 9	6.15 D	*INHBB*	Inhibin subunit beta B	10.0 D
*CPM*	Carboxypeptidase M	6.23 U	*ACTA2*	Actin, alpha 2, smooth muscle, aorta	4.66 D
*ITIH5*	Inter-alpha-trypsin inhibitor heavy chain family member 5	6.58 D	*LOC100505633*	Long intergenic non-protein coding RNA 1133	5.19 D
*AP1S3*	Adaptor-related protein complex 1 subunit sigma 3	11.3 D	*FOXQ1*	Forkhead box Q1	4.96 U

^1^ Fold change shows the up (U)- or down (D)- regulated fold change of the respective gene on SDECM. Adjusted *p*-values were ≤ 1.8 × 10^−50^. Genes highlighted with the same colour were common between the two comparisons.

**Table 4 cells-08-00085-t004:** Upregulated **^1^** transcription factors in the *COL1A1* gene promoter/enhancer.

Gene Symbol	Gene Name	TCP FC	TCP padj	SDECM FC	SDECM padj
*BRCA1*	BRCA1, DNA repair associated	4.60	3.9 × 10^−38^	6.54	2.1 × 10^−31^
*CEBPA*	CCAAT enhancer binding protein alpha	2.11	2.7 × 10^−1^	4.82	8.0 × 10^−4^
*E2F8*	E2F transcription factor 8	8.16	1.1 × 10^−22^	21.21	3.2 × 10^−22^
*EZH2*	Enhancer of zeste 2 polycomb repressive complex 2 subunit	5.76	1.5 × 10^−46^	3.87	1.1 × 10^−18^
*FOSL1*	FOS-like 1, AP-1 transcription factor subunit	4.82	5.0 × 10^−43^	1.10	6.5 × 10^−1^
*MXD3*	MAX dimerization protein 3	4.79	1.8 ×10^−24^	4.43	2.7 × 10^−11^

^1^ Upregulated at P4 by ≥4-fold on either TCP or SDECM. Highlighted cells indicate significant data. FC—fold change, padj—adjusted *p*-value.

**Table 5 cells-08-00085-t005:** Stable genes on SDECM (P1 vs. P4) that were differentially expressed on TCP ^1^.

Gene Symbol	Gene Name	Fold Change	padj
*FOSL1*	FOS Like 1, AP-1 transcription factor subunit	4.81 U	5.0 × 10^−43^
*STC1*	Stanniocalcin 1	4.73 D	2.2 × 10^−40^
*CDH2*	Cadherin 2	11.7 U	1.0 × 10^−27^
*NXPH3*	Neurexophilin 3	4.46 D	7.1 × 10^−19^
*POM121L9P*	POM121 transmembrane nucleoporin-like 9, pseudogene	10.9 D	9.6 × 10^−19^
*MSC*	Musculin	4.62 U	4.3 × 10^−12^
*POSTN*	Periostin	7.64 U	1.2 × 10^−11^
*CCRL1*	Atypical chemokine receptor 4	8.45 D	4.7 × 10^−10^
*ADAMTS14*	ADAM metallopeptidase with thrombospondin type 1 motif 14	4.60 U	8.6 × 10^−10^
*ACSS1*	Acyl-CoA synthetase short chain family member 1	4.24 D	1.5 × 10^−8^
*SH3TC2*	SH3 domain and tetratricopeptide repeats 2	5.20 U	1.1 × 10^−6^
*NTF3*	Neurotrophin 3	4.00 U	2.1 × 10^−6^
*MYBPH*	Myosin binding protein H	6.36 D	1.2 × 10^−5^
*ELOVL2*	ELOVL fatty acid elongase 2	9.81 U	1.2 × 10^−4^
*GRIK5*	Glutamate ionotropic receptor kainate type subunit 5	4.46 D	9.2 × 10^−3^

^1^ Differentially regulated at P4 by ≥4-fold on TCP.

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
