# Peer review of "Transcriptome-Wide Analysis of Human Chondrocyte Expansion on Synoviocyte Matrix"

_cells, 2019, doi:10.3390/cells8020085_

Round 1

Reviewer 1 Report

In this paper, the Authors analyse the difference between chondrocytes cultured on tissue culture plastic (TCP), (morphological and phenotypical changes with loss of the ability to produce hyaline-like matrix) and chondrocytes grown on synoviocyte-derived extracellular matrix (SDECM) (that reduces this de-differentiation allowing for more than twice the number of population doublings whilst retaining chondrogenic capacity).

Specifically they apply RNA-Seq analysis to examine the differences between TC plastic-expanded and SDECM-expanded human chondrocytes. The data presented in this paper indicate as SDECM provides a chondrogenesis-preserving environment for cell expansion and potential mechanisms for the enhancement of chondrogenesis-preserving growth.

I recommend dealing with the following implementations.

Major comments and concerns:

1)   The differentiation of plastic-cultured cells is a general problem. It has been described for example that hepatocytes rescue their ability to differentiate in short time when cultured on hydrogel (Cozzolino et al., 2016). This aspect should be better discussed for different cell types.

2)   Regarding the genes related to cartilage development (but also ECM and motility), the Authors show that some of them are decreased on TCP respect to SDECM. Since this seems to be the central point of the paper, the Authors should show the validation of these seq data both at RNA and protein levels (at least RNA).

Minor comments:

Some typing errors should be amended.

Author Response

Major comments and concerns:

1)   The differentiation of plastic-cultured cells is a general problem. It has been described for example that hepatocytes rescue their ability to differentiate in short time when cultured on hydrogel (Cozzolino et al., 2016). This aspect should be better discussed for different cell types.

The authors have expanded upon the description of chondrocyte loss of re-differentiation potential with population doublings in the introduction “The use of the term de-differentiation is distinct from the use of the term in re-programming or stem cell literature, as it indicates that the cells no longer have the ability to form hyaline like cartilage tissue.”. In addition, further discussion of the effect of culture surface in re-differentiation has been included in the discussion section page 10 lines 289-295.

2)   Regarding the genes related to cartilage development (but also ECM and motility), the Authors show that some of them are decreased on TCP respect to SDECM. Since this seems to be the central point of the paper, the Authors should show the validation of these seq data both at RNA and protein levels (at least RNA).

The authors agree that validation of genes is important for further/deeper study. This study is an expansion of previous work (Kean & Dennis 2015) which showed typical chondrocyte-related gene expression changes by RT-qPCR (COL1A1, COL2A1, SOX9, ACAN, MATN1, MMP13, COL10A1) the levels of which, by FPKM, are well correlated. In addition, RNA-Seq has been shown to be well correlated with RT-qPCR data across over 15,300 genes (Everaert 2017). This text has been included in the introduction.

Minor comments:

Some typing errors should be amended.

The manuscript has been thoroughly reviewed for typing errors.

Reviewer 2 Report

1. For understanding the molecular mechanism regarding the difference between TCP and SDECM, The authors should analyze extracellular stimuli or clarify relationship between expression of ECM components and chondrocyte specific transcription factors. To provide insight into the mechanism, further functional gene ontology, gene-network, and pathway analysis should be performed.

2. Figure 4, Due to relatively low expression of the transcription factors in Passage 1 SDECM, it seems that expression of the transcription factors was decreased in Passage 4 SDECM, especially for Sox6, Col1A1. Please discuss about this.

3. Please provide deeper and more analytic discussion about Table 5.

Author Response

1. For understanding the molecular mechanism regarding the difference between TCP and SDECM, The authors should analyze extracellular stimuli or clarify relationship between expression of ECM components and chondrocyte specific transcription factors. To provide insight into the mechanism, further functional gene ontology, gene-network, and pathway analysis should be performed.

More analyses have been performed and included. A simplified figure of cell cycle related genes on SDECM and an analysis of ECM organization related gene differential expression across all comparisons.  

2. Figure 4, Due to relatively low expression of the transcription factors in Passage 1 SDECM, it seems that expression of the transcription factors was decreased in Passage 4 SDECM, especially for Sox6, Col1A1. Please discuss about this.

SOX6 was 3.2-fold decreased in passage 4 on SDECM padj 1.5x10-8, which is beneath the 4-fold cut off that was set for gene analysis, which is why it is not highlighted as being differentially expressed in the figure. COL1A1 is not in the cartilage development gene list, however, increased expression of COL1A1 is related to fibrocartilage development and is often shown as an indicator of de-differentiation: COL1A1 expression was 1.2-fold upregulated at P4 on SDECM padj =0.13, its expression on TCP was 2.8-fold upregulated at P4 on TCP padj 1x10-30. These details were included in the manuscript on page 3 lines163-166. COL11A1, which was in the gene list, was significantly decreased on both TCP and SDECM as indicated to the left of Figure 4, this has been clarified by increasing the figure size and filling of the arrows.

3. Please provide deeper and more analytic discussion about Table 5.

Further discussion of table 5 and analyses has been included.

Reviewer 3 Report

The paper entitled "Transcriptome-wide analysis of human chondrocyte expansion on synoviocyte matrix" is well written and potentially very interesting to anyone using or studying chondrocytes in vitro and cartilage tissue engineering.

My main concern is the difficulty for the reader to extract meaningful results from the huge amount of data. For instance, how do the differences in gene expression explain the increase in PDs on SDECM? Showing genes that promote proliferation or inhibit proliferation in Figure 3 would be helpful. A similar type of simplified graph for cells cultured on SDECM would be welcome.

Similarly, Figure 4 is difficult to read; showing genes that promote or inhibit chondrogenesis on TCP or SDECM would be helpful to understand what kinds of genes are responsible for the chondrogenesis-preserving environment of the SDECM. The discussion could therefore be greatly improved by discussing some of the genes responsible for the increased proliferation and chondrogenicity of SDECM.

Furthermore, the conclusion mentions that culture on SDECM affects genes related to motility; however, this seems to be completely omitted from the discussion.

Here are also a few minor comments:

P3, l93-95 : I suppose PD means population doubling ?

P8, l157 : « green=up and red=down » : on the heatmap, the green arrow seems to be pointing down and the red arrow up; is that correct?

P10, l225: I’m not sure what “futile” means in this context

P11, l238: “in” is written twice in a row

Author Response

My main concern is the difficulty for the reader to extract meaningful results from the huge amount of data. For instance, how do the differences in gene expression explain the increase in PDs on SDECM? Showing genes that promote proliferation or inhibit proliferation in Figure 3 would be helpful. A similar type of simplified graph for cells cultured on SDECM would be welcome.

A simplified graph for the cells cultured on SDECM has been prepared as well as amendments to the current figure. One of the issues with many of the genes in this graph is that they can be both associated with promoting or inhibiting proliferation; this is now stated in the text. Also, we had neglected to include in the legend that the arrow colors signified that the genes identified in that box were related to promoting, inhibiting or just associated with proliferation. These terms/colors have now been included.

Similarly, Figure 4 is difficult to read; showing genes that promote or inhibit chondrogenesis on TCP or SDECM would be helpful to understand what kinds of genes are responsible for the chondrogenesis-preserving environment of the SDECM. The discussion could therefore be greatly improved by discussing some of the genes responsible for the increased proliferation and chondrogenicity of SDECM.

The authors were unsure of any restrictions to the size or number of figures, it doesn’t seem that there are any. This figure has been increased in size and the arrows filled for clarity. Part of the problem/issue is that we don’t yet know which genes promote or inhibit chondrogenesis. This screen identified far too many genes for our small lab to investigate in depth individually. It is also complicated by the fact that SDECM does not completely repress de-differentiation, meaning that whilst 15 of the cartilage related genes were unchanged on SDECM whilst differentially expressed on TCP. It would be presumptive to state that these are chondrogenic capacity maintaining. More discussion of the proliferation and maintenance of chondrogenic capacity have been included.

Furthermore, the conclusion mentions that culture on SDECM affects genes related to motility; however, this seems to be completely omitted from the discussion. 

Further discussion of the effect on motility has been added.

Here are also a few minor comments:

P3, l93-95 : I suppose PD means population doubling ?

Yes, it has now been defined in both the abstract and here, at the first mention in the text.

P8, l157 : « green=up and red=down » : on the heatmap, the green arrow seems to be pointing down and the red arrow up; is that correct?

There were two errors in this figure which have been corrected. To better align with the heatmap colors and preserve red/green colorblind accessibility, Red=up and blue=down, the arrows have also been filled and moved to the right of the gene symbol as this makes them clearer.

P10, l225: I’m not sure what “futile” means in this context

This sentence has been modified to “This potentially futile growth, a conversion from reversible cell cycle arrest to irreversible senescence, could be indicative of geroconversion as described by Blagosklonny [14-16].” for clarification.

P11, l238: “in” is written twice in a row

This has been corrected, thank you.

Round 2

Reviewer 1 Report

The paper can be accepted in the present form.

Author Response

The authors would like to thank the reviewers for their comments and appreciate that the manuscript is much improved because of them.

Reviewer 3 Report

The manuscript has been significantly improved, especially concerning the figures that are much easier to read. I still have a few comments and suggestions:

Figures 3 and 4:

On TCP, 59 genes are associated with promoting cell proliferation and 140 with inhibiting it. On SDECM, 68 genes are associated with promoting cell proliferation and 51 with inhibiting it. This seems to suggest why cells grow more on SDECM; is this reasonable? If so, the authors could briefly mention this in the results section or in the discussion.

I sometimes find it difficult, when looking at the heatmaps, to understand how certain gene regulations were found: for instance, the heatmap in Fig6 suggests that BMP2 is up-regulated in comparison 3 and SOX9 is up-regulated in comparison 3 and especially 4; on the other hand, I don’t see how COL2A1 in down-regulated in comparison 4. Is this correct? As SDECM is supposed to “provide a more chondrogenesis-preserving environment”, I am surprised by the results shown on the right part of figure 6 (the arrows)…

L261-262: “300 transcriptions factors identified in the COL1A1 gene”: this expression does not sound appropriate; do you mean “300 transcription factors with an identified (or potential) binding sequence in the COL1A1 gene”?

L347-348: “An increase in… unexpanded cells”: this sentence is a bit difficult to understand and should be rephrased. (the increase is “deficient” in unexpanded cells?)

L377-378: “potentially an effect… such as cancer”: this sentence is also very difficult to understand and should be rephrased.

Author Response

The authors would like to thank the reviewers for their comments and appreciate that the manuscript is much improved because of them.

In response to your last comments:

Figures 3 and 4:

On TCP, 59 genes are associated with promoting cell proliferation and 140 with inhibiting it. On SDECM, 68 genes are associated with promoting cell proliferation and 51 with inhibiting it. This seems to suggest why cells grow more on SDECM; is this reasonable? If so, the authors could briefly mention this in the results section or in the discussion.

That is partly our interpretation as well, although we had expected to see this in comparison 3 at P1 on SDECM vs. TCP. Looking at that data, cell proliferation comes up in the GO term list for overall differentially expressed genes but not in the up or down regulated analyses, indicating that some of those genes went up and some went down for it to be identified as enriched in the overall gene list. Looking at which of those genes went up or down doesn’t clarify the issue and leads us to speculate that growth factors such as CTGF, FGFs and IGFs may be present in the SDECM as their expression went down. The following has been added to the discussion: “The increased number of genes promoting proliferation on SDECM vs. TCP (Figs 3 & 4), and perhaps more importantly, the ratio of genes promoting vs. inhibiting proliferation is thought to be partially responsible for the increased proliferation of chondrocytes on SDECM.”

I sometimes find it difficult, when looking at the heatmaps, to understand how certain gene regulations were found: for instance, the heatmap in Fig6 suggests that BMP2 is up-regulated in comparison 3 and SOX9 is up-regulated in comparison 3 and especially 4; on the other hand, I don’t see how COL2A1 in down-regulated in comparison 4. Is this correct? As SDECM is supposed to “provide a more chondrogenesis-preserving environment”, I am surprised by the results shown on the right part of figure 6 (the arrows)…

It is difficult, and we need to remember that this is just a snapshot in time of proliferating chondrocytes, not differentiating chondrocytes. There may have been a file conversion problem though, as in Fig. 6 BMP2 is significantly downregulated in comparisons 1 and 2 and there is no significant change in comparisons 3 and 4. I do see the color difference in the BMP2, SOX9 and no color difference with COL2A1 and looking at the supplemental data, I see that BMP2 was upregulated on SDECM by 3-fold with a p-value of 0.028, meaning that it failed to achieve the parameters delineated in the determination of ‘significant’. In comparison 4, SOX9 was upregulated 2.7-fold with a p-value of 1.2x10-22 and COL2A1 was 20-fold downregulated with a p-value of 8.9x10-4. Heatmaps were generated from FPKM values and this might partly explain the lack of concordance. That has now been specified in the methods. To clarify that no arrow means that the comparison was not significant, the text has been amended in the figure legends to read: “The arrows indicate which comparisons were significantly different at a ≥4-fold change with p < 0.01; no arrow means that the comparison did not meet that threshold (1=TCP P1 vs. P4, 2=SDECM P1 vs. P4, 3=TCP vs. SDECM at P1, 4=TCP vs. SDECM at P4).”

L261-262: “300 transcriptions factors identified in the COL1A1 gene”: this expression does not sound appropriate; do you mean “300 transcription factors with an identified (or potential) binding sequence in the COL1A1 gene”?

Yes, the text has been amended to: Out of the over 300 transcription factors with an identified (or potential) binding sequence in the COL1A1 gene [12] only 5 were significantly upregulated on TCP and 4 on SDECM Table 4.”

L347-348: “An increase in… unexpanded cells”: this sentence is a bit difficult to understand and should be rephrased. (the increase is “deficient” in unexpanded cells?)

 “An increase in chondrogenic markers is often achieved but is deficient compared to unexpanded cells [19, 20].” Has been rephrased to: “An increase in chondrogenic markers is often achieved in cell culture expanded cells [19, 20]. However, the expression of these markers is commonly deficient when compared to non-expanded chondrocytes.”

L377-378: “potentially an effect… such as cancer”: this sentence is also very difficult to understand and should be rephrased.

“Network and gene enrichment analyses of the genes in Table 5 showed no overall connection between them; potentially an effect of the dearth of studies on musculoskeletal transcriptomics in comparison to more widely studied topics such as cancer.” Has been rephrased to “Network and gene enrichment analyses of the genes in Table 5 showed no overall connection between them. This is potentially a consequence of the dearth of studies on the musculoskeletal transcriptome, particularly in cartilage, whereas many cancer studies have contributed to the gene ontology terms and their association.”